# VAL1 acts as an assembly platform co-ordinating co-transcriptional repression and chromatin regulation at Arabidopsis *FLC*

Pawel Mikulski [1,4] ✉, Philip Wolff [1], Tiancong Lu [1,5], Mathias Nielsen[1], Elsa Franco Echevarria [2], Danling Zhu[1,6], Julia I. Questa [1,7], Gerhard Saalbach[3], Carlo Martins[3] & Caroline Dean [1,2] ✉

Polycomb (PcG) silencing is crucial for development, but how targets are specified remains incompletely understood. The cold-induced Polycomb Repressive Complex 2 (PRC2) silencing of *Arabidopsis thaliana FLOWERING LOCUS C* (*FLC*) provides an excellent system to elucidate PcG regulation. Association of the DNA binding protein VAL1 to *FLC* PcG nucleation region is an important step. VAL1 co-immunoprecipitates APOPTOSIS AND SPLICING ASSOCIATED PROTEIN (ASAP) complex and PRC1. Here, we show that ASAP and PRC1 are necessary for co-transcriptional repression and chromatin regulation at *FLC*. ASAP mutants affect *FLC* transcription in warm conditions, but the rate of *FLC* silencing in the cold is unaffected. PRC1-mediated H2Aub accumulation increases at the *FLC* nucleation region during cold, but unlike the PRC2-delivered H3K27me3, does not spread across the locus. H2Aub thus involved in the transition to epigenetic silencing at *FLC*, facilitating H3K27me3 accumulation and long-term epigenetic memory. Overall, our work highlights the importance of VAL1 as an assembly platform co-ordinating activities necessary for epigenetic silencing at *FLC*.

From the elegant genetic and molecular analysis of *Drosophila*, the Polycomb group (PcG) proteins have been shown to play a central role in developmental gene regulation[1]. They function in distinct complexes to maintain epigenetic silencing of genomic targets[2]. The most well-studied complexes are Polycomb Repressive Complex 1 (PRC1), which monoubiquitinates lysine (118/119/121-dependent on the organism) on histone H2A (H2Aub), and Polycomb Repressive Complex 2 (PRC2), which methylates histone H3 tails at lysine 27 (H3K27me). Initially thought to be sequentially recruited to the same targets, many studies have now demonstrated a much more complex scenario with variant and canonical PRC1[3,4], interdependent

recruitment with PRC2 and/or co-operative interactions mediating association to Polycomb Response Elements (PREs)[5–9]. There is also a lack of clarity about what specifies target sites as Polycomb targets and how transcription and chromatin silencing feedback influence each other[10]. Further dissection of the molecular events underpinning Polycomb silencing is, therefore, necessary to define the core mechanistic principles.

Arabidopsis *FLOWERING LOCUS C* (*FLC*) is a PcG target central to the process of vernalization, ensuring flowering in favorable conditions[11,12]. *FLC* encodes a floral repressor whose expression is increased in winter annual Arabidopsis accessions through the activity

[1]Cell and Developmental Biology, John Innes Centre, Norwich, UK. [2]MRC Laboratory of Molecular Biology, Cambridge, UK. [3]Biological Chemistry, John Innes Centre, Norwich, UK. [4]Present address: Department of Biochemistry, University of Oxford, Oxford, UK. [5]Present address: State Key Laboratory of Plant Genomics and National Center for Plant Gene Research, Institute of Genetics and Developmental Biology, Chinese Academy of Sciences, Beijing, China. [6]Present address: SUSTech-PKU Institute of Plant and Food Science, Department of Biology, Southern University of Science and Technology, Shenzhen, China. [7]Present address: Centre for Research in Agricultural Genomics, Barcelona, Spain. ✉e-mail: pawel.mikulski@bioch.ox.ac.uk; caroline.dean@jic.ac.uk

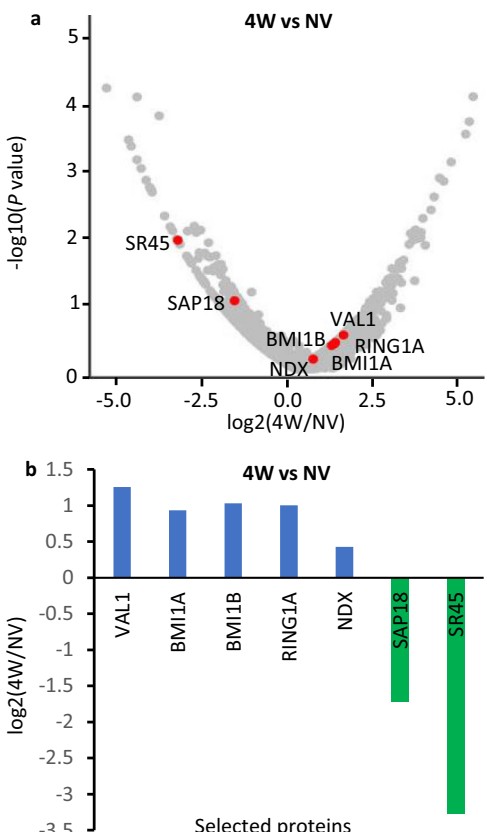

**Fig. 1 | VAL1 proteomic interactors as identified by CoIP-MS. a** Volcano plot showing relative protein abundance in log2 ratio of 4 weeks cold (4W) to non-vernalized (NV). Proteins of interest are labelled and highlighted in red. **b** Bar plot showing relative protein abundance in log2 ratio of 4 weeks cold (4W) to non-vernalized (NV) for selected proteins. Blue bars show proteins with increased abundance upon transfer to cold, green bars show proteins with decreased abundance upon transferred to cold.

of FRIGIDA[13]. Short cold exposure in autumn/winter results in a relatively rapid transcriptional repression of *FLC*, in a process involving FRIGIDA sequestration[14] and cold-induced *FLC* antisense transcripts called *COOLAIR*[15]. Over the longer timescale of winter, once transcriptionally repressed, *FLC* is epigenetically silenced through a PRC2-dependent epigenetic switch[16]. This switch occurs at individual *FLC* alleles with a low probability[17], leading to progressive silencing over the whole plant. A cold-induced step in this switch is the VIN3/VRN5-dependent nucleation of H3K27me3 at an intragenic site that covers three nucleosomes between the transcription start site and the beginning of intron 1[18–20]. When plants return to warm temperatures, H3K27me3 spreads to cover the entire gene, a feature required for long-term epigenetic silencing throughout the rest of development[17]. The *FLC* system thus provides the opportunity to dissect the components required for a PRC2 epigenetic switch.

Previously, we had shown that the transcriptional repressor VIVIPAROUS1/ABI3-LIKE (VAL1) is a factor acting early in the switching mechanism necessary for H3K27me3 nucleation at *FLC*[21]. VAL1 recognizes conserved RY motifs in *FLC* intron 1 with a one bp mutation in the first RY motif sufficient to completely block the vernalization process. The *val1* mutation attenuated *FLC* cold-induced transcriptional repression, but not the long-term Polycomb memory. VAL1 was shown to interact with HDAC and PRC1 components and members of APOPTOSIS AND SPLICING ASSOCIATED PROTEIN (ASAP), which interacted with the PRC2 accessory proteins VIN3 and VRN5[21]. This then provided a putative connection between the sequence at the nucleation region and the machinery

for transcriptional repression and long-term PcG silencing. A functional link between VAL1, PRC1, and HDAC factors with PRC2 silencing has been extensively described[22–27]. But how they all connected to silence *FLC* during vernalization remained unclear.

Recent work using a trans-factor tethering system elegantly showed how VAL1 can recruit PRC1 and PRC2 and couple these to HDAC activity[28]. VAL1 could directly recruit the PRC1 component BMI1, whereas PRC2 recruitment was proposed to be mediated by interaction with SAP18, TPL/TPR, or PRC1 through the EAR domain of VAL1[28]. It was thus of interest to address how these factors functioned in the temporal sequence involved in the silencing of the endogenous *FLC* locus, in a genotype sensitive to vernalization. Using proteomic analysis, RNA PolII occupancy, chromatin modifications, and chromatin-associated RNA enrichment assays, we show that ASAP and PRC1 work in a network of co-transcriptional regulators that effectively coordinate different steps in *FLC* silencing.

## Results

### ASAP-VAL1 and PRC1-VAL1 complexes may affect different phases of *FLC* silencing

Previous proteomics analyses demonstrated the in vivo interaction of VAL1 with components of the ASAP and PRC1 complexes[21]. However, which components were direct interactors and in what phase of *FLC* silencing they acted had remained unclear. We extended the proteomic analysis using VAL1-HA as bait on plant samples and compared before (non-vernalized), after 4 weeks cold exposure and then after 4 weeks cold and 7 days subsequent warm (Fig. 1a, b and Supplementary Fig. 1a–c and Supplementary Table 1). The general conclusion from these data was that VAL1 preferentially associates with components of the PRC1 complex (BMI1A, BMI1B and RING1A). The occurrence of these interactions increased after cold exposure along with increase of VAL1 (Fig. 1a, b, Supplementary Table 1). We also observed an interaction between VAL1 and ASAP complex components (SR45 and SAP18) (Fig. 1a, b, Supplementary Table 1) as previously shown[21], albeit with low abundance of these components. In contrast to PRC1, ASAP abundance decreased upon transfer to cold (Fig. 1a, b). The differential abundance of ASAP and PRC1 peptides within and between treatments suggested non-stoichiometric representation in a VAL1 complex, or multiple independent complexes of various functional importance at different phases of *FLC* silencing. A surprise finding was that VAL1 immunoprecipitated NDX, a homeodomain protein we have previously described as stabilizing an R-loop at the 3′ end of *FLC*[29]. Our analysis to determine which were direct interactors of VAL1 was complicated by difficulties in expressing VAL1-tagged proteins in *N. benthamiana*. By a yeast two-hybrid (Y2H) analysis, we found that VAL1 weakly interacts with SAP18 (Supplementary Fig. 2a), but this interaction was not sufficiently strong to be detected in co-immunoprecipitation experiments in transfected HEK293 cells (Supplementary Fig. 2b). NDX interacts indirectly via PRC1 components RING1A and RING1B (Supplementary Fig. 2c, d)[30].

### Mutations of the ASAP complex affect *FLC* expression in warm but not cold-treated plants

To investigate when the ASAP function was most important for *FLC* silencing, we crossed ASAP mutants into a *FRI*+ genotype in order to confer a vernalization requirement. The generated *FRI* genotypes were analysed for *FLC* expression over a vernalization time course (Fig. 2a). Plants were harvested before any cold exposure (non-vernalized; NV), immediately after 2, 4, or 6 weeks of cold treatment (2 W, 4 W, 6 W) and after 6 weeks of cold followed by a warm period for 7 days (6 WT7). In a parallel experiment, we also performed *FLC* expression analysis in vernalization-independent *fri* background (Fig. 2b). Consistent with the changed interactions found in the proteomics experiments, the major effect of the ASAP mutants in both *FRI* and *fri* genotypes was in warm (NV) conditions (Fig. 2a, b

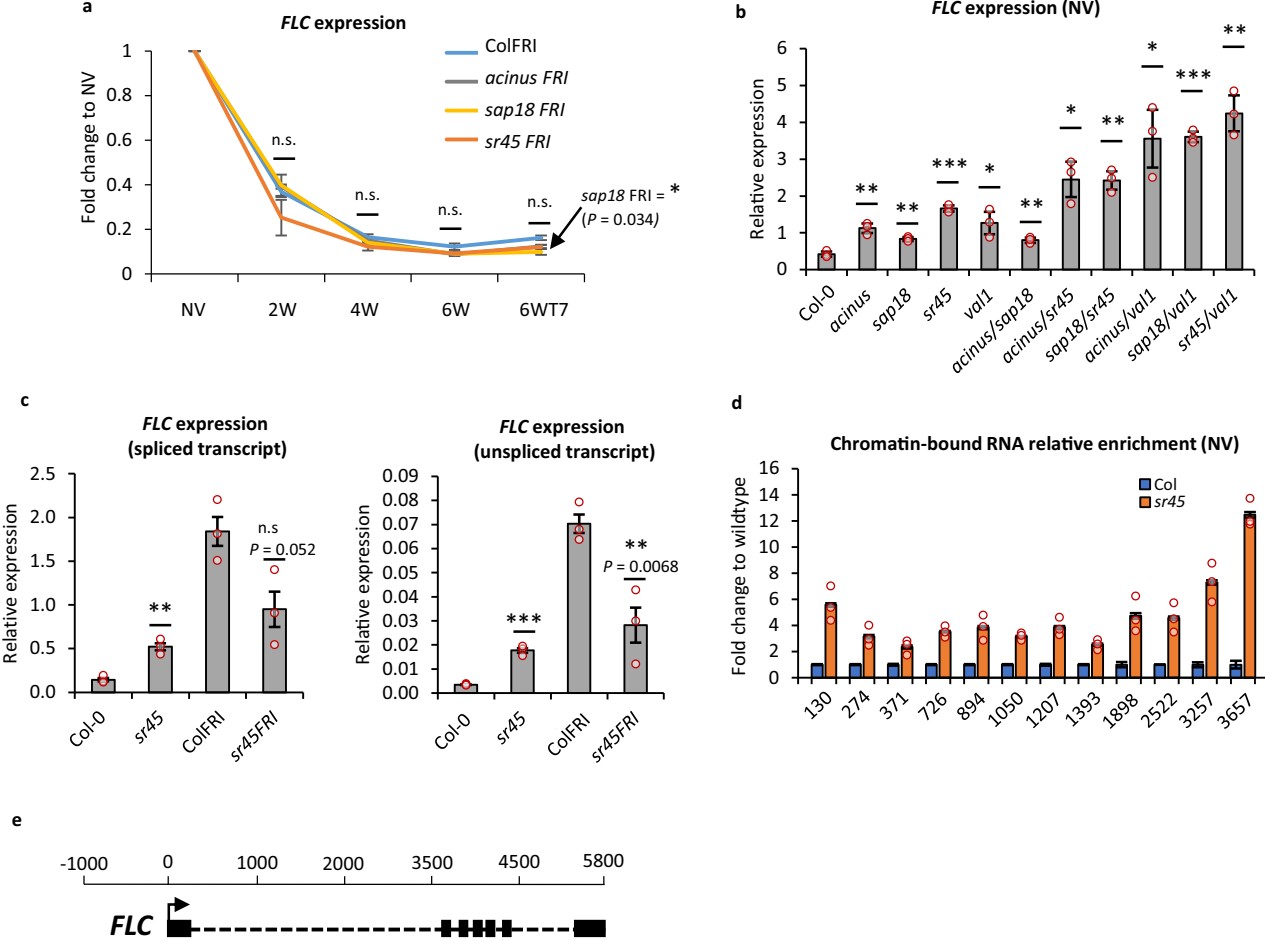

**Fig. 2 | ASAP mutants affect *FLC* transcription in warm conditions. a** *FLC* downregulation (spliced transcript) during vernalization (fold change to NV). Values were normalized as relative expression to *UBC*. *N* = 3 biological replicates, error bars = propagated SEM. NV non-vernalized, 2/4/6 W 2/4/6 weeks cold, 6 WT7 6 weeks cold followed by 7 days in warm. *P* value for *sap18 FRI* at 6 WT7 0.0344. **b** *FLC* expression (spliced transcript) in ASAP single mutants and different double mutant combinations combining different ASAP mutants or ASAP mutants with *val1*. Values correspond to the mean relative expression to *UBC*. *N* = 3 biological replicates; error bars = SEM. **c** *FLC* expression in *sr45* in a vernalization-dependent (FRI) and vernalization-independent genotype (Col-0). Values correspond to the mean relative expression to *UBC*. Significance tests compare Col-0 vs *sr45* and ColFRI vs *sr45FRI*. *N* = 3 biological replicates, error bars = SEM. **d** Chromatin-bound *FLC* RNA enrichment over the locus. Data were shown as fold change to wildtype. X-axis shows the midpoint of the amplicon relative to *FLC* transcriptional start site (TSS). *N* = 3 biological replicates, error bars = propagated SEM. Statistics are calculated with two-tail Student's *t*-test in comparison to wildtype (*$P \leq 0.05$; **$P \leq 0.01$; ***$P \leq 0.001$). **e** *FLC* gene model schematic. The scale bar indicates the position relative to the transcriptional start site (TSS), where TSS = 0.

and Supplementary Fig. 3a–c). The ASAP mutants did not influence the rate of *FLC* silencing after a different cold and post-cold treatments−the rate of *FLC* downregulation was similar in ASAP mutants and in the wildtype (Fig. 2a). Furthermore, our flowering time analyses showed that *FLC* upregulation observed in ASAP mutants in warm does not result in significantly different flowering time comparing to the wildtype (except for *acinus*) (Supplementary Fig. 3a). This reinforces the view that ASAP functions at *FLC* at a specific temporal phase in the vernalization process and/or that other genes compensate for its loss during cold exposure.

To understand the relationship of VAL1 and ASAP function, we generated double mutant combinations in a *fri* background and analysed in NV conditions. We observed a strong increase in *FLC* expression in warm-grown double mutants combining *val1* and ASAP mutants (Fig. 2b), consistent with VAL1 interacting with many complexes to effect *FLC* silencing. The strongest release of *FLC* silencing in warm conditions was seen in combinations containing *val1*, *sr45*, and *sap18* mutations (Fig. 2b). As expected with the general correlation of sense and antisense transcription at *FLC*[31], *COOLAIR* was also upregulated in single ASAP mutants in warm conditions (Supplementary Fig. 3b).

An interesting genetic interaction was observed between *sr45* plants, with and without an active *FRIGIDA* allele (Fig. 2c). *sr45* increased *FLC* levels in a *fri* background but decreased *FLC* levels relative to wildtype when combined with *FRI*+. This genetic interaction could be accounted for if FRI functions through SR45 to de-repress ASAP function. Previous studies show FRI and ASAP working in co-transcriptional regulation linking nascent RNA processing to RNA PolII functioning with feedbacks to chromatin modification[14,32,33]. This, together with the lack of an ASAP mutant phenotype during cold and post-cold stages suggests that the major role of ASAP at the *FLC* locus is in quantitative co-transcriptional repression that sets the level of *FLC* expression in the warm.

## SR45-dependent co-transcriptional repression at *FLC*
The broad roles of the ASAP complex at the other targets[34] led us to pursue the link between co-transcriptional and chromatin regulation. Since *sr45* showed the strongest effect on *FLC* expression, we profiled the co-transcriptional and chromatin changes at *FLC* comparing *sr45* mutant and wildtype−Col-0 (*fri*) or ColFRI (*FRI*+). *sr45* increased relative *FLC* transcript levels, not only in total RNA (Fig. 2c), but also

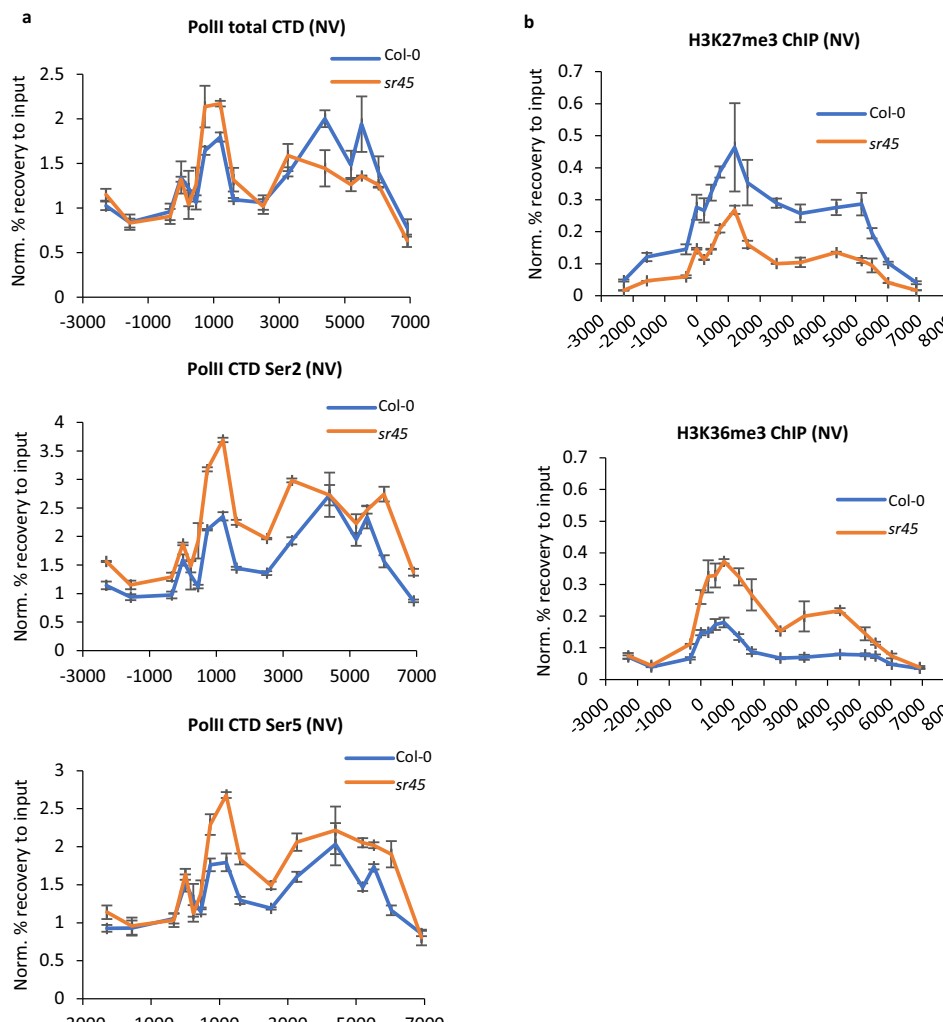

**Fig. 3 | SR45 influences the profile of RNA PolII and histone modifications at** ***FLC.*** **a** RNA Polymerase II (PolII) ChIP enrichment. X-axis depicts the midpoint position of the amplicon at the *FLC* locus, amplicon 0 = transcriptional start site (TSS). Y-axis corresponds to % recovery to input values normalized to the house-keeping gene *ACT7*. NV non-vernalized; $N = 3$ biological replicates; error bars = SEM.

**b**. SR45-dependent changes in active/repressive histone modifications. X-axis shows the midpoint of the amplicon at *FLC*, amplicon 0 = TSS. Y-axis corresponds to % recovery to input values normalized to H3 enrichment. NV non-vernalized; $N = 3$ biological replicates; error bars = SEM.

chromatin-associated nascent RNA (Fig. 2d, e) and *COOLAIR* increased co-ordinately (Supplementary Fig. 3b). Given the connection of SR45 to splicing we analysed splicing efficiency at *FLC* introns 1–3 but found no changes caused by loss of SR45 or other ASAP components in steady state RNA (Supplementary Fig. 3d). We also checked RNA PolII occupancy between *sr45* and Col-0 by chromatin-IP (ChIP) and found some variation in total RNA PolII occupancy at *FLC* in specific regions of *FLC* (Fig. 3a). In addition, levels of RNA PolII phosphorylated on serine-2 and serine-5 both increased over *FLC* in *sr45* (Fig. 3a). RNA PolII serine-2 phosphorylation is associated with active transcription and consistent with its increase we found the balance of active and repressive chromatin modifications changed in warm-grown *sr45* plants, with increased H3K36me3 and decreased H3K27me3 at *FLC* (Fig. 3b). SR45-dependent co-transcriptional repression is thus important for establishing the chromatin state at the locus as plants grow in the warm, with FRI potentially antagonizing that co-transcriptional step.

## VAL1 affects specific nucleosome dynamics

Next, we asked whether VAL1 played a role in these co-transcriptional processes in addition to its function in Polycomb nucleation[21]. We measured chromatin-associated *FLC* RNA in warm-grown *val1*

seedlings as a readout for *bona fide FLC* transcription. Like the relative increase in total RNA[21], and like *sr45* (Fig. 2d), *val1* increased chromatin-associated *FLC* RNA (Fig. 4a). Given the VAL1 binding site is down-stream of the *FLC* promoter, we asked if such transcriptional changes might be accompanied by changes in nucleosome dynamics[35]. We adapted the histone salt fractionation protocol, that preferentially extracts nucleosomes recently disrupted by either transcription or torsional stress[35] and used it to analyze nucleosomes over *FLC*. The fraction of low-salt extractable nucleosomes decreased at the +1 and +2 nucleosomes in cold-treated samples (Supplementary Fig. 4), agreeing with the previous observation of cold-induced nucleosome stabilization[19]. Furthermore, we observed that *val1* increased the low-salt extractable fraction for nucleosomes +1 and +3, but not for nucleosome +2 (Fig. 4b, c). Nucleosomes +1 and +3 correspond to the edges of the H3K27me3-enriched nucleation region found in *FLC* after cold exposure[36]. These VAL1-dependent changes were confined to chromatin extracted from warm-grown *val1* seedings and were not found in cold-treated samples (Supplementary Fig. 4). Interestingly, they were not connected to gross changes in chromatin accessibility as judged by FAIRE (Formaldehyde-Assisted Isolation of Regulatory Elements)[37] (Fig. 4d). Thus, our data suggest that VAL1 affects local nucleosome stability without a concomitant change in general

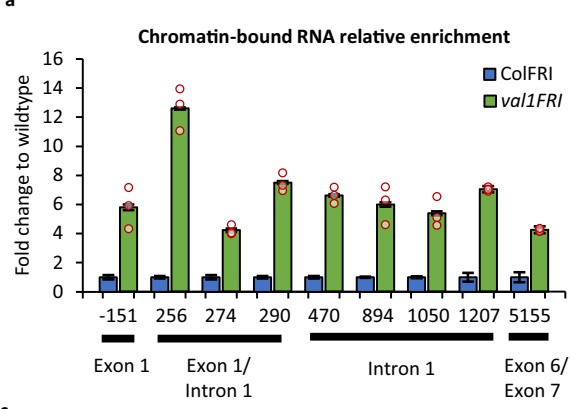

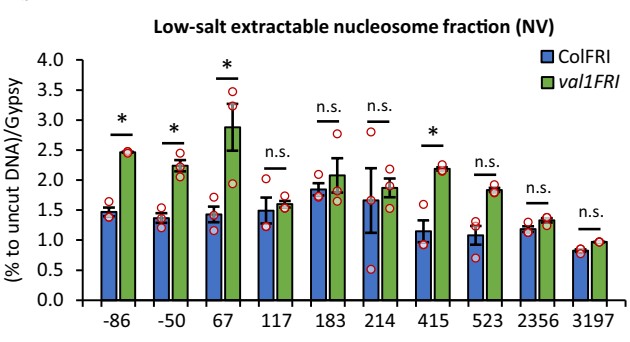

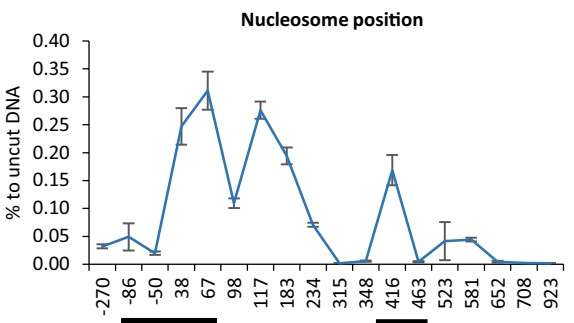

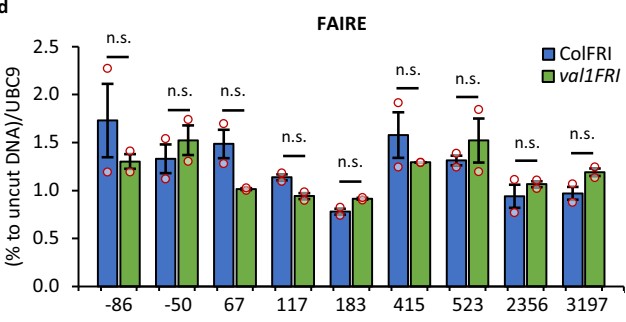

**Fig. 4 | VAL1 regulation of *FLC* chromatin. a** Chromatin-bound *FLC* RNA enrichment over the *FLC* locus is shown as a fold change to wildtype. X-axis shows the midpoint of the amplicon relative to *FLC* transcriptional start site (TSS), amplicon 0 = TSS. *N* = 3 biological replicates, error bars = propagated SEM. **b** Low salt extractable nucleosome fraction over *FLC* shown as % recovery to uncut DNA and relative to AT4G07700 (Gypsy-like transposon). X-axis shows the beginning of the amplicon relative to TSS, amplicon 0 = TSS. *N* = 3 biological replicates; error bars = SEM; asterisk = *P* < 0.05 from two-tail Student *t*-test in comparison wildtype vs *val1*; n.s. non-significant; NV non-vernalized. **c** Nucleosome position at *FLC* in ColFRI wildtype. Results from MNase-qPCR assay shown as % to uncut DNA. X-axis shows the beginning of the amplicon relative to *FLC* TSS, amplicon 0 = TSS. *N* = 2 biologically independent samples (separate seedlings' plates), error bars = SEM. Lines below the axis indicate regions of increased low-salt nucleosome extractability in *val1* shown in **b**. **d** Chromatin accessibility at *FLC* measured as FAIRE enrichment. Shown as % recovery to non-crosslinked sample (UN-FAIRE) and relative to *UBC9*. X-axis shows the beginning of amplicon relative to TSS, amplicon 0 = TSS. Statistics are calculated from the Student *t*-test in comparison wildtype vs *val1* for all amplicons individually. *N* = 2 biologically independent samples (separate seedlings' plates); error bars = SEM; NV non-vernalized.

chromatin accessibility and that the +1 and +3 nucleosome *FLC* regions are potentially occupied by nucleosomes with different salt solubility, potentially under different torsional stress[38,39]. Overall, these data connect VAL1-induced co-transcriptional regulation to nucleosome dynamics at the edges of the *FLC* nucleation region, a region where RNA PolII occupancy is high.

## H2Aub accumulates during cold at the *FLC* nucleation region but does not spread like H3K27me3 upon return to warm

Having characterized the major role of VAL1-ASAP in the warm, we then investigated the functional significance of the VAL1-PRC1 in *FLC* silencing. Our proteomics analysis showed VAL1-PRC1 in vivo interaction both before and during cold (Fig. 1b). As H2Aub is the prominent histone mark deposited by PRC1, we undertook the analysis of H2Aub accumulation at *FLC* through a vernalization time course. The vernalization-sensitive genotype ColFRI showed relatively low levels of H2Aub accumulation at the nucleation region after growth in the warm (Fig. 5a). As *FLC* expression quantitatively decreased in plants exposed to cold, H2Aub levels increased (Fig. 5a). However, H2Aub levels at *FLC* increased only up to 4 weeks (Fig. 5a), and then reduced by 6 weeks (Supplementary Fig. 5). This temporary accumulation during the cold exposure is consistent with a need for H2Aub during epigenetic switching, but then a requirement for its removal to stabilize the accumulated H3K27me3[40]. H2Aub has recently been shown to associate with chromatin accessibility at transcriptional regulation hotspots[41] rather than with transcriptional

repression per se, potentially explaining the different dynamics between H2Aub accumulation and PRC2-induced H3K27me3 enrichment at the *FLC*. Analysis of non-vernalized lines with or without FRI showed H2Aub levels at the low expressing, high H3K27me3 *fri FLC* locus were much higher than in *FRI*+, where the locus is actively transcribed and associated with H3K36me3 (Fig. 5b). H2Aub enrichment was also detected at the 3' end of *FLC* in *fri* plants (+5600 *FLC* amplicon; Fig. 5b)−an important regulatory region of *FLC* as it forms a gene loop with the *FLC* 5' nucleation region[42,43]. Overall, these results suggest that PRC1-delivered H2Aub is low in *FRI*+ plants before vernalization as *FLC* is actively transcribed, accumulates progressively with cold at the nucleation region, until H3K27me3 levels are relatively high, and then decreases.

We then tested how H2Aub changed relative to H3K27me3 in warm conditions post-cold, when *FLC* chromatin transitions from a metastable silenced state through to a long-term silencing associated with the spreading of H3K27me3 across the locus[17]. As expected, H3K27me3 accumulated in the nucleation region during cold exposure (6 W) and spread across the whole gene body after transfer back to warm for 10, 14, and 20 days (6WT10/14/20) (Fig. 5c). H2Aub accumulated in the nucleation region up to 4 weeks cold, like the H3K27me3 and increased for 10 (6WT10) and 14 (6WT14) days after transfer back to the warm but did not spread across the locus (Fig. 5c). Interestingly, 20 (6WT20) days after transfer back to the warm, the H2Aub peak at the nucleation decreased back to levels seen at the end of 6 W cold (6 W). A slow,

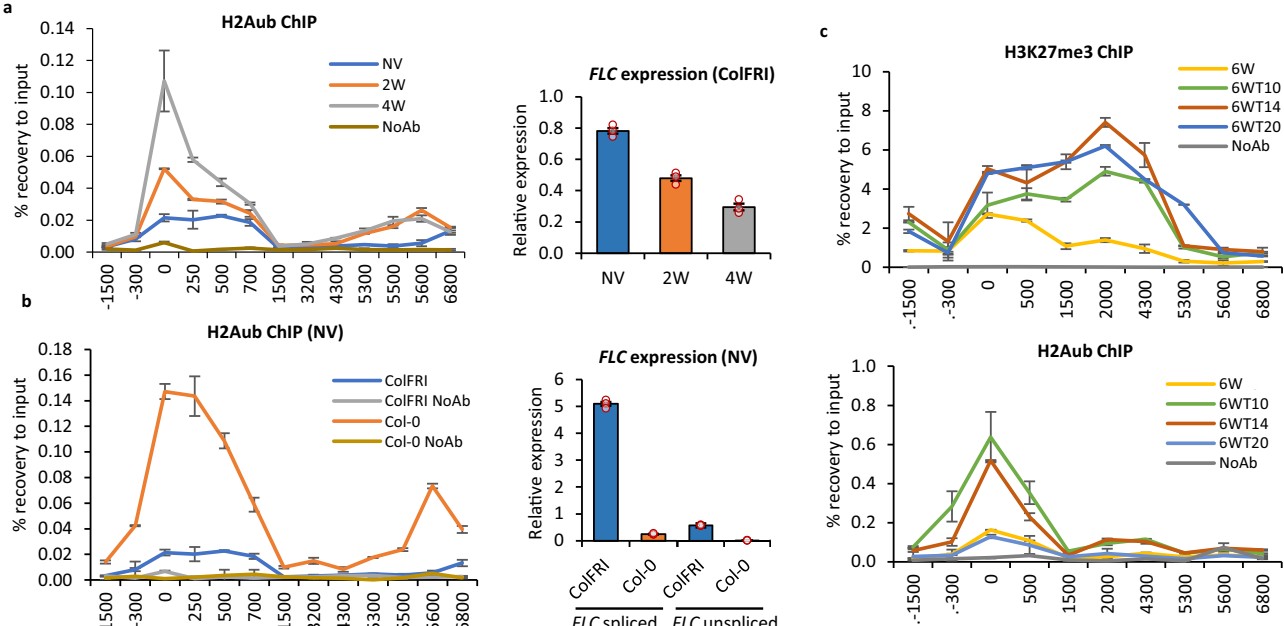

**Fig. 5 | H2Aub dynamics at *FLC*. a** H2Aub ChIP showing increased enrichment of H2Aub at the 5′ end of *FLC* in ColFRI during vernalization correlating with decreasing transcription. The graph on the right shows unspliced *FLC* expression during a vernalization time course measured by qRT-PCR. Values correspond to the mean relative expression to *UBC*. **b** H2Aub ChIP enrichment showing anticorrelation of *FLC* expression with H2Aub levels. *FLC* expression in wildtype backgrounds in NV is depicted in the graph on the right. **c** Histone mark behavior during post-cold. H3K27me3 and H2Aub enrichment over *FLC*. **a**–**c** X-axis shows the midpoint of the amplicon relative to *FLC* TSS (amplicon 0 = TSS). *N* = 3 biological replicates; error bars = SEM; NoAb no antibody negative control. NV non-vernalized; 2/4/6 W 2/4/6 weeks of cold treatment; 6WT10/14/20 6 weeks cold followed by 10/14/20 days post-cold.

gradual decrease post-cold and lack of spreading match VRN5 occupancy at *FLC*[17]. H2Aub enrichment at *FLC* was unaffected in *vrn5* (and in a core PRC2 component mutant *vrn2*) in non-vernalized conditions and only slightly reduced after 6 weeks cold exposure (Supplementary Fig. 6). The lack of difference in NV conditions agrees with H2Aub being generally independent of LHP1 and PRC2 in Arabidopsis[44] and suggests that VRN5 is not necessary for H2Aub accumulation, but rather its enrichment could be dependent on preceding H2Aub enrichment at the *FLC* nucleation region.

Despite the low levels of H2Aub accumulation at *FLC* chromatin in the warm, and the clear increase of H2Aub in *FLC* region during cold (which correlates with PRC2 assembly and early spreading), *FLC* expression levels in *prc1* single mutants are like wildtype in Col-0 (*fri*), likely due to genetic redundancy of PRC1 components[22,23,27]. However, how FRIGIDA would affect this phenotype had not been investigated before. We, therefore, combined *ring1A* and *bmi1B* mutants with *val1* in a *FRI* + background and analysed *FLC* and *COOLAIR* expression during a vernalization time course. The rates of *FLC* downregulation and *COOLAIR* upregulation during cold, as well as *FLC* and *COOLAIR* reactivation post-cold, were not substantially different between double mutants: *val1 ring1A* or *val1 bmi1B* and *val1* single mutant in *FRI* + background (Supplementary Figs. 7a–d, 8a, b). In addition, we did not observe differences in flowering time after vernalization in PRC1 mutants in *FRI* + genotypes (Supplementary Fig. 8c), aside from a slight delay in *ring1A FRI* as reported previously for *ring1A* in Col-0 background[45]. Higher levels of *COOLAIR* proximal transcript were found in *val1 bmi1B FRI* double mutant compared to either single mutant after 4 W cold (Supplementary Fig. 7d), potentially suggesting a decoupling of the PRC1 effect on *FLC* and *COOLAIR* at later stages of vernalization. Overall, our data show that the PRC1 mark H2Aub dynamically accumulates at the *FLC* nucleation region preceding the accumulation of H3K27me3 during cold exposure.

## VAL1 and NDX physically interact and are required for H2Aub accumulation at the *FLC* nucleation region

To pursue the functional importance of the VAL1-NDX-PRC1 interactions, we undertook a genetic analysis between *val1*, PRC1 mutants, and *ndx* (*ndx1-4*). This revealed that NDX is required for the *FLC* upregulation caused by *val1*, and that loss of PRC1 does not affect the consequences of *NDX* deficiency (Fig. 6a). The highest *COOLAIR* upregulation was in *ndx*, with the degree of upregulation being dependent on PRC1 and VAL1 (Supplementary Fig. 9a). Supporting interactions within the VAL1-NDX-PRC1 pathway, we also showed that NDX, VAL1, or VAL1-recognized RY motifs are necessary for full H2Aub enrichment at *FLC* (Fig. 6b, c), as well as H3K27me3 (Supplementary Fig. 9b and as published in ref. 21). Thus, the VAL1-NDX interplay would seem to be important in the co-transcriptional regulation required for H2Aub accumulation and the switch to epigenetic silencing. Given the role of NDX in the stabilization of the *COOLAIR*-induced R-loop[29,46], it will be important to establish if these genetic interactions point to an additional R-loop role in the *FLC* nucleation region or are the result of a gene loop physically linking the 3′ and 5′ end of the gene[42].

## Discussion

How transcriptional repression links to epigenetic silencing is far from understood. Cell type-specific or developmentally induced transcriptional repressors bind different Polycomb complexes but direct recruitment models are too simplistic[9,10]; linear genetic hierarchies have not been established and functional output of PREs has been shown to be heavily dependent on the chromatin context[47]. Our findings that VAL1 acts as an assembly platform co-ordinating ASAP and PRC1 activities that occur independently in *FLC* regulation help explain some of this complexity. It also raises interesting parallels to the non-linear recruitment activities of the PcG proteins at Drosophila PREs[7,48,49], with ASAP co-transcriptional repressors and PRC1 components likely redundantly establishing the chromatin environment required for robust PRC2 silencing (Fig. 7).

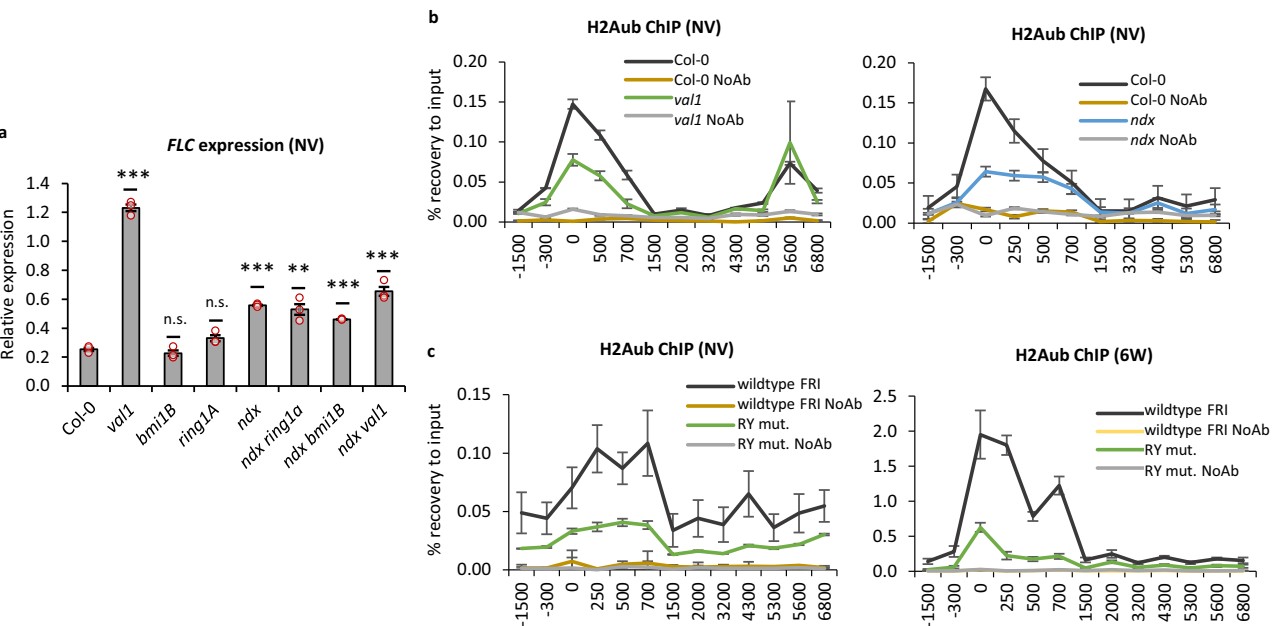

**Fig. 6 | VAL1, PRC1, and NDX are required for H2Aub accumulation at the *FLC* nucleation region.** *FLC* expression and H2Aub enrichment depend on VAL1, NDX, and PRC1. **a** *FLC* expression in mutants of VAL1, PRC1, and NDX. Values correspond to the mean relative expression to the geometric mean of *UBC* and *PP2A*. Statistics are calculated with a two-tail Student's *t*-test in comparison to Col-0. *N* = 3

biological replicates; error bars = SEM. **b**, **c** H2Aub ChIP results. X-axis shows the midpoint of the amplicons relative to the *FLC* Transcriptional Start Site (amplicon 0 = TSS). NoAb no antibody negative control; *N* = 3 biological replicates for H2Aub IPs and three qPCR technical replicates for NoAb controls; error bars = SEM. NV non-vernalized, 6 W 6 weeks (cold).

ASAP and PRC1 have been documented to have very different roles in gene regulation in different organisms. ASAP has previously been shown to play a role as part of the spliceosome and exon junction complexes in many organisms[34,50], whereas PRC1 functionality is associated with H2Aub and transcriptional silencing[51]. In plants, ASAP has been associated with the regulation of innate immunity[52] and abiotic responses[53] and considered to function through altering RNA metabolism or alternative splicing[32,54]. However, we did not detect any effect of an *sr45* mutation on alternative splicing or splicing efficiency of the *FLC* sense strand introns. Instead, we found changes linked to co-transcriptional repression, namely RNA PolII post-translational modifications and histone modifications associated with Polycomb target ON or OFF states. We thus propose that SR45, as an ASAP component, influences a co-transcriptional step modulating RNA PolII complex composition and thus the chromatin modifications deposited during transcription. The ASAP complex clearly regulates *FLC* during growth in the warm, as judged by mutant phenotypes, but if it functions in cold, it does so redundantly with other factors.

The PRC1-mediated H2Aub modification has been associated with transcriptional regulation and epigenetic stability in Drosophila and mammalian cells[54]. At *FLC*, the PRC1-H2Aub is associated with transcriptional repression in warm conditions in Col-0, or in response to cold in *FRI* + genotypes. In this respect, PRC1 appears to function at *FLC* more as a variant PRC1 rather than a canonical PRC1. The timescale of the VAL1-PRC1-mediated H2Aub accumulation at the *FLC* nucleation region during cold exposure suggests an intermediary role between the transcriptional silencing determined by ASAP-VAL1 and the long-term PRC2 epigenetically stable spread state. However, the genetic redundancy makes PRC1 functionality hard to tease apart so we were unable to precisely dissect the interdependency of PRC1-delivered H2Aub, ASAP function and PRC2-delivered H3K27me3 in the epigenetic switch at *FLC*. The co-transcriptional changes linked to VAL1-dependent H2Aub deposition thus appear to be important steps, which work together with

multivalent trans-factor interactions to robustly recruit PcG complexes to their target sites.

VAL1 has been reported to interact with many proteins. In addition to components of ASAP and PRC1, it has been shown to interact with LHP1[25], histone deacetylases (HDA9 and HDA6)[55–57], and Mediator complex component (MED13)[55]. The new interaction found here between VAL1 and the R-loop stabilizer NDX[29] points to a potential relationship with R-loops. It will be important in the future to understand what mediates these interactions and how their association is epigenetically inherited at the nucleation region, like at Drosophila PREs. It will also be interesting to continue to dissect whether the ASAP complex only has a function in warm conditions, or its function is covered in the early cold by other factors. These are likely to involve the antisense transcripts at *FLC* (*COOLAIR*) that contribute to transcriptional silencing during cold exposure, which associate with the chromatin at the 5′ end of *FLC*, and in which natural variation has a functional consequence[43,50].

Overall, we interpret our data as showing that VAL1 functions as an assembly platform to coordinate co-transcriptional repression and chromatin regulation at Arabidopsis *FLC*. We envisage that transcription is downregulated by multiple repressor complexes (ASAP and PRC1) either linked to RNA PolII and interacting with VAL1 at the nucleation region or assembled at the nucleation region. These appear to function independently in the warm, with H2Aub playing a major role in the cold associated with *FLC* silencing, alongside PRC2 deposition of H3K27me3. Given the parallels with the Drosophila PcG mechanism, the activities of these many protein complexes are likely to function cooperatively to influence the different steps in the mechanism. Further elucidation will require a detailed analysis of the co-transcriptional silencing, chromatin modification, and nucleosome remodeling.

## Methods
### Plant material
All mutants and transgenic lines were either in Col-0 or FRI^sf2 (ColFRI) background[21,51]. Unless otherwise stated, experiments in non-vernalized

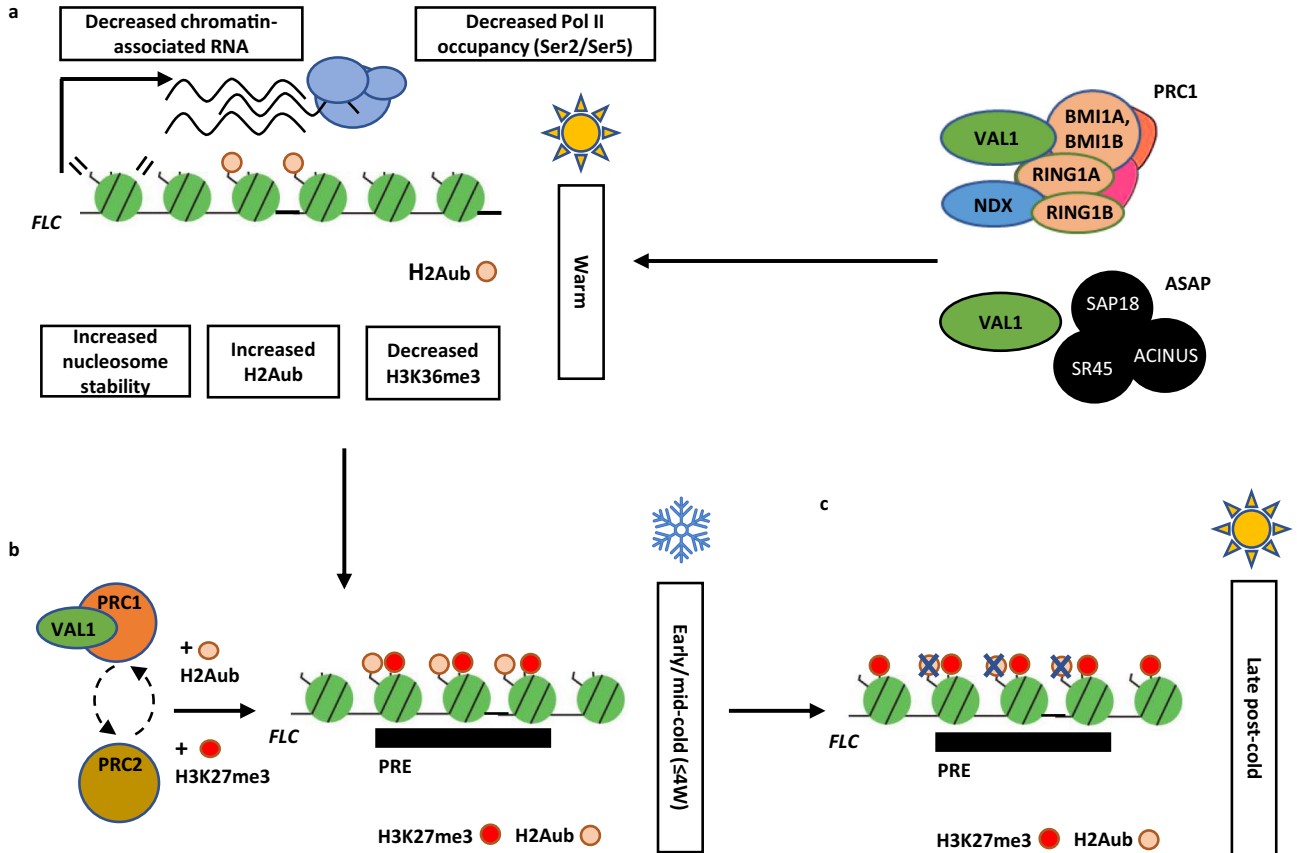

**Fig. 7 | Functional relationship of ASAP, PRC1, and PRC2 (H3K27me3) in *FLC* regulation. a** ASAP and PRC1 act as co-repressors that regulate *FLC* chromatin features and downregulate *FLC* expression in warm. **b** Upon shift to cold, PRC1-mediated H2Aub accumulates at *FLC* PRE along with H3K27me3 through the coordinated action of VAL1, PRC1, and PRC2. H2Aub and H3K27me3 are present at

*FLC* PRE in an increasing proportion of cells during cold. 4 W 4 weeks cold. **c** Subsequently, in late post-cold (return to warm) timepoints, H2Aub is lost from the *FLC* PRE, whereas H3K27me3 enrichment increases and spreads over the *FLC* gene body, where it is stably maintained long-term. H2K27me3 thus marks long-term epigenetic silencing with H2Aub serving as a transient repression signal.

conditions used Col-0 as a wildtype. The analyses in the vernalization time course used ColFRI as a wildtype to capture dynamics of cold-induced *FLC* expression and chromatin changes. Mutant alleles were used as described previously: *bmi1B* (*drip1-1*), *bmi1A* (*drip2-1*), *ring1A* (GK-293A04), *ring1B* (SALK_117958), *ndx1-4*, *flc* (*flc-2*), RY motif (*vrn8*), *val1−2*, *vin3-4*, *vrn2-1*, *vrn5-8*, *sap18* (SALK_02363C), *sr45* (SALK_018273), and *acinus* (SALK_078554)[17,21,29,30,58]. Double mutants were generated by crosses between homozygous single mutant lines and selected by PCR-based genotyping using primers listed in Supplementary Table 2.

### Growth conditions
Seeds were surface sterilized and sown on MS-GLU (MS without glucose) media plates and stratified at 4 °C in the dark for 2 days. For non-vernalized (NV) conditions, seedlings were grown for 14 days in long-day conditions (16 h light, 8 h darkness with a constant 20 °C). For vernalization treatment, seedlings were grown at 5 °C in short-day conditions (8 h light, 16 h darkness with constant 5 °C) for the number of weeks indicated in the respective experiment description.

### Protein co-immunoprecipitation in Arabidopsis
Four grams of Arabidopsis seedlings were crosslinked and ground to a fine powder in liquid N2, following ChIP protocol[21]. Immunoprecipitation of protein complexes was done as described in[17] using Protein A beads (Thermo Fisher Scientific, #10008D) coupled with HA antibody (CST, #3724). IP product was resuspended in 2X NuPAGE sample buffer (Thermo Fisher Scientific, #NP0007) and denatured at 70 °C for 15 min with 100 mM DTT. The final samples were run on NuPAGE Bis-Tris gels (Thermo Fisher Scientific, # NP0321) and subjected to in-gel digestion.

Protein bands corresponding to heavy and light chains of antibody were excised from NuPAGE gels. The rest of gels were combined and prepared according to standard procedures adapted from[59]. Briefly, gel slices were washed with 50 mM Triethylammonium bicarbonate (TEAB) buffer pH 8 (Sigma, #T7408), incubated with 10 mM DTT for 30 min at 65°C, followed by incubation with 30 mM iodoacetamide (IAA) at room temperature (both in 50 mM TEAB) for alkylation of cysteine residues. After washing and dehydration with acetonitrile, the gels were soaked with 50 mM TEAB containing 10 ng/µl Sequencing Grade Trypsin (Promega, #V5111) and incubated at 50°C for 8 h. The peptides were eluted with an equal volume of 5% formic acid followed by different steps of acetonitrile concentration (up to 50%). The combined supernatants were dried in a SpeedVac concentrator (Thermo Fisher Scientific, #SPD120) for mass spectrometry, and the peptides were dissolved in 0.1%TFA/3% acetonitrile.

### Protein co-immunoprecipitation in mammalian cells
HEK293T cells were seeded on poly-L-lysine coated plates at ~70% confluency and transfected with a DNA:PEI (1:3.5) mixture after cells had attached, one 89 mm tissue culture dishes per CoIP was used. After attachment, cells were transfected with 8 µg total DNA (2 µg of GFP-VAL1 and 6 µg of Flag-dsRed-SAP19, BMI1B, or TPL). Cells were lysed ~18 h post-transfection in lysis buffer (20 mM Tris pH 7.4, 200 mM NaCl, 10% glycerol, 5 mM NaF, 2 mM Na3VO4, 1 mM EDTA, 0.2% Triton X-100, EDTA-free protease inhibitor cocktail (Roche)). Lysates were clarified by centrifugation (16100 rcf, 10 min), and supernatants were incubated with GFP-trap (Chromotek) for 90 min at 4 °C on an overhead tumbler. Immunoprecipitates were washed three times in lysis

buffer and eluted by boiling in LDS sample buffer for 10 min. Input and CoIP fractions were separated by SDS-PAGE, blotted onto PVDF membrane, checked for equal loading by Ponceau staining, and processed for Western blotting with appropriate antibodies. Primary antibodies (anti-GFP (Sigma, #G1544) or anti-FLAG (Sigma, #7425)) and secondary antibodies were diluted 1:5000 in PBS, 0.1% Triton X-100 and 5% milk powder. Blots were washed with PBS, 0.1% Triton X-100 and developed with ECL Western Blotting Detection Reagent on film.

## Mass spectrometry

Samples were analyzed by nanoLC-MS/MS on an Orbitrap Fusion Tribrid mass spectrometer coupled to an UltiMate 3000 RSLCnano LC system (Thermo Fisher Scientific, # IQLAAEGAAPFADBMBCX). The samples were loaded and trapped using a pre-column with 0.05% TFA at 20 μl/min for 3 min. The trap column was then switched in-line with the analytical column (nanoEase M/Z column, HSS C18 T3, 100 Å, 1.8 μm; Waters, #186008818) for separation using the following gradient of solvents: A (water, 0.05% formic acid) and B (80% acetonitrile, 0.05% formic acid) at a flow rate of 0.3 μl min-1: 0-3 min 3% B (trap only); 3-4 min linear increase B to 6%; 4-71 min increase B to 50%; 71-76 min increase B to 65%; followed by a ramp to 99% B and re-equilibration to 3% B in 23 min. Data were acquired with the following mass spectrometer settings in positive ion mode: MS1/OT: resolution 120K, profile mode, mass range m/z 300-1800, AGC 4e5, fill time 50 ms; MS2/IT: data dependent analysis was performed using parallel CID and HCD fragmentation with the following parameters: top20 or top25 in IT rapid mode, centroid mode, quadrupole isolation window 1.6 Da, charge states 2-7, threshold 1.5e4, CE = 30, AGC target 1e4, max. inject time 35 ms, dynamic exclusion 1 count, 15 s exclusion, exclusion mass window ±5 ppm or 2 counts, 30 s, 7 ppm.

The raw data was processed in Proteome Discoverer 3.0 (Thermo Scientific, Waltham, USA). Spectra were recalibrated and filtered for top20 peaks per 100 Da. Identification was performed using the Chimerys search node with the inferys_2.1_fragmentation prediction model. Parameters were enzyme trypsin, 2 missed cleavages, oxidation (M) as variable and carbamidomethylation (C) as fixed modification, 0.6 Da fragment tolerance. Evaluation was performed using Percolator. For peak detection and quantification, the Minora Feature Detector was used with a min. trace length of 7 and S/N 3. After normalisation to total peptide amount the quantification was based on the top3 unique peptides per protein. Missing values were imputed by low abundance resampling. For hypothesis testing a background-based T-test was applied. Maximum fold change was set to 100. The counts from VAL1-HA[21] bait samples were calculated in comparison to Col-HA negative controls (ColFRI containing empty construct with HA tag). Results were exported to Microsoft Excel and filtered to remove contaminants and proteins with less than 5 unique peptides.

## Yeast two-hybrid analysis

Yeast two-hybrid protocol was adapted from ref. 60. The AH109 strain was used for transformation, following the protocol described in the Yeast Protocols Handbook (version no. PR973283 21; Clontech), with both Gal4-BD and Gal4-AD constructs added. In detail, yeast cultures of wildtype strain AH109 were grown up to OD600 = 0.8 at 28 °C on Yeast Peptone Dextrose (YPD). The cells were spun at 2000 rcf and washed twice with Buffer 1 (0.1 M LiAc, 1 M Sorbitol, 0.5x TE buffer pH 7.5). Final cell pellet was resuspended in Buffer 2 (0.1 M LiAc, 40% PEG 3350, 1x TE buffer pH 7.5) and incubated for 10 min at room temperature. About 0.5–1 μg plasmid DNA containing Gal4-BD and Gal4-AD constructs and 10 μg salmon sperm was mixed with 600 μL of Buffer 2 and added to the cells, followed by incubation at 30 °C for 30 min. 30 μL DMSO was added, the mixture was vortexed and incubated for 20 min at 42 °C. The mixture was spun for 3 min at 4000 rcf, resuspended in 200 μL Synthetic Dextrose (SD) medium and plated onto Petri dishes

with solid SD medium lacking Trp and Leu (SD-LW) for transformant selection. Transformed cells were grown for 3 days. Transformed yeast were grown to OD600 = 0.8 and spotted onto selective SD medium additionally lacking His (SD-LWH) which permitted the identification of weak protein interactions. Yeast dilutions during spotting were: 1/1, 1/5, 1/25, 1/25.

## Expression analysis

Total RNA extraction was performed using the hot phenol method, as described elsewhere[61]. Genomic DNA contamination was removed using a TURBO DNA-free kit (Invitrogen, #AM1907) following the manufacturer's guidelines. cDNA was synthesized using SuperScript IV First-strand Synthesis System (Invitrogen, #18091050), using gene-specific primers. cDNA was diluted ten times before qPCR. All primers are listed in Supplementary Table 2. cDNA was amplified using SYBR Green I Master (Roche, 04887352001) and run on LightCycler 480 machine (Roche, 05015243001). Ct values were normalized to the geometric mean of reference genes: *UBIQUITIN CARRIER PROTEIN 1 (UBC)* and/or *SERINE/THREONINE PROTEIN PHOSPHATASE 2 A (PP2A)*. Chromatin-associated RNA has been isolated using urea as published before[62].

## Nucleosome stability analysis

The protocol has been adapted from ref. 63. About 2 g Arabidopsis seedlings were ground in liquid nitrogen, resuspended in 35 ml NIB buffer (10 mM MES-KOH pH 5.4, 10 mM NaCl, 10 mM KCl, 2.5 mM EDTA, 250 mM Sucrose, 0.1 mM Spermine, 0.5 mM Spermidine, 1 mM DTT, 1x Protease inhibitor cocktail EDTA-free (Thermo Fisher Scientific, # A32965)) and filtered through one layer of miracloth. 20% Triton X-100 was added to final concentration of 0.35%, the extract was rotated at 4 °C for 10 min, and spun at 1000×g for 10 min. The pellet was transferred to Low Protein Binding tubes (Thermo Fisher Scientific, #90410) and washed twice with MNase buffer (10 mM Tris-HCl pH 8, 5 mM NaCl, 2.5 mM CaCl2, 2 mM MgCl2, 1x Protease inhibitor cocktail EDTA-free (Thermo Fisher Scientific, # A32965)) at 1000×g for 3 min. The pellet was weighted and MNase buffer was added in a ratio of 5 μL buffer:1 μg pellet. The mixture was prewarmed at 37 °C for 5 min. About 150 μL of the mixture was put aside as an uncut sample and the other 150 μL was subjected to 5 U/ml (final concentration) MNase (Takara Bio, #2910 A) enzymatic digestion at 37 °C for 10 min. The concentration of MNase was experimentally tested to obtain a nucleosome ladder as in ref. 63. The reaction was stopped by adding EGTA to final concentration of 2 mM. The mixture was spun at 1000×g for 3 min, supernatant was removed, the pellet resuspended in 80 mM salt buffer (80 mM NaCl) and rotated at 4 °C for 15 min. The samples were spun at 1000×g for 3 min and the supernatant was collected as a mobile nucleosome fraction. Samples were adjusted to 500 μL and subjected to DNA recovery by adding 1/50 volume 0.5 M EDTA, 1/50 vol. 5 M NaCl and 2 μL RNAse A (Thermo Fisher Scientific, #EN0531). The samples were incubated at 37 °C for 30 min. Proteins were removed by adding 1/16 vol. 10% SDS to a final concentration of 0.63% and 2.5 μL Proteinase K (Ambion, #AM2546), followed by incubation at 50 °C for 1 h. The samples were then subjected to standard phenol-chloroform-isoamyl alcohol and DNA precipitation with ethanol. The enrichment was calculated by qPCR with primers indicated in Supplementary Table 2. Ct values in mobile nucleosome fraction were normalized to uncut DNA and Gypsy-like transposon (At4g07700) as an internal control, following normalizations as published in ref. 37.

## Micrococcal nuclease assay (MNase-qPCR)

About 2 g Arabidopsis seedlings were ground in liquid nitrogen and the pellet was resuspended in Honda buffer[21], filtered through miracloth, and spun at 1000×g, 4 °C for 10 min, similarly to ChIP protocol. Subsequently, the pellet was washed twice in MNase buffer (10 mM Tris-HCl pH 8, 5 mM NaCl, 2.5 mM CaCl2, 2 mM MgCl2, 1x Protease

inhibitor cocktail EDTA-free (Thermo Fisher Scientific, # A32965)) and finally resuspended in 1 mL MNase buffer. The samples were divided into 300 μL uncut control and 300 μL MNase-digested sample. The mixtures were prewarmed at 37 °C for 5 min and digested samples were treated with 66.6 U/mL (final conc.) MNase enzyme (Takara Bio, #2910 A) at 37 °C for 30 min. The reactions were stopped by adding 1/100 volume 0.5 M EDTA and subjected to protein extraction by adding 0.05% (final conc.) SDS, followed by standard phenol-chloroform-isoamyl alcohol and DNA precipitation with isopropanol. The enrichment was calculated by qPCR with primers indicated in Supplementary Table 2. Ct values in mobile nucleosome fraction were normalized to uncut DNA.

**Formaldehyde-assisted isolation of regulatory elements (FAIRE)**
FAIRE protocol was adapted from ref. 37. The samples were separated into FAIRE (fixed) and UN-FAIRE (unfixed) samples. Preparation of UN-FAIRE samples follows the same protocol as FAIRE samples, but without the Formaldehyde crosslinking step. In detail, 1 μg seedlings were washed three times in Phosphate-buffered saline (PBS) and fixed in 1% Formaldehyde under vacuum for 10 min. Fixation was quenched by adding 0.125 M Glycine under vacuum for 5 min. The seedlings were rinsed three times in PBS and dried on a paper towel. The tissue was ground to a fine powder using a mortar and pestle for 10–15 min. The powder was resuspended in ice-cold Buffer 1 (400 mM sucrose, 10 mM Tris-HCl pH 8.0, 5 mM Beta-mercaptoethanol, 0.1 mM PMSF, one tablet of Complete Protease Inhibitor Cocktail per 50 mL of buffer) and incubated at 4 °C (cold room) on a rotor for 10 min. Homogenized suspension was filtered through two layers of Miracloth and spun at 2880 rcf for 20 min. The pellet was resuspended in 1 mL Buffer 2 (50 mM sucrose, 10 mM Tris-HCl, pH 8.0, 10 mM MgCl$_2$, 1% Triton X-100, 5 mM Beta-mercaptoethanol, 0.1 mM PMSF, ½ Complete Protease Inhibitor Tablet per 10 mL of the buffer) and transferred to 1.5 mL Eppendorf tube. The samples were spun at 12000 rcf for 10 min at 4 °C. Wash with Buffer 2 was repeated two more times. The pellet was resuspended in 300 μL Buffer 3 (1.7 M Sucrose, 10 mM Tris-HCl pH 8, 0.15% Triton X-100, 2 mM MgCl$_2$, 5 mM Beta-mercaptoethanol, 1 mM PMSF, 1/2 Complete Protease Inhibitor Tablet per 10 mL buffer). The resuspended pellet was overlaid onto 300 μL Buffer 3 and spun at 16000 rcf for 60 min at 4 °C. The pellet was resuspended in Buffer 4 (50 mM Tris-HCl pH 8.0, 10 mM EDTA, 1% SDS, 0.1 mM PMSF, ¼ Complete Protease Inhibitor Tablet per 5 mL of the buffer) by vortexing. The mixture was sonicated in three rounds of sonication for 5 min each (30 s ON/30 s OFF cycles) and mixed in between the rounds. The optimal resulting fragment size should be 0.2–0.8 kb. The mixture was spun at 16000 rcf for 10 min at 4 °C and supernatant was transferred to a new tube. Equal (300 μL) volume of Phenol-Chloroform-Isoamyl alcohol (PCI) solution was added and vortexed for 5 min. The solution was spun at 12000 rcf for 10 min at room temperature. The upper aqueous phase was transferred to a new tube. 0.1 volume of 3 M sodium acetate, 2.5 volume of absolute pure ethanol, and 1 μL glycogen was added. The solution was incubated at −80 °C for 1 h, then spun at 16000 rcf for 45 min at 4 °C. The pellet was washed with 70% ethanol twice and spun at 11000 rcf for 10 min at room temperature. The pellet was dried up for 5–10 min at room temperature and resuspended in DNAse-free water. The enrichment was assessed by qPCR using primers indicated in Supplementary Table 2. Calculations were performed using the ΔΔCt method as described in ref. 37 with normalization of a crosslinked sample (FAIRE) to non-crosslinked sample (UN-FAIRE) and then to *UBC9* (AT4G27960) as an internal control.

**Chromatin immunoprecipitation**
Histone modification and PolII ChIP was performed as previously described[17,62], respectively. Immunoprecipitation was done with antibodies: anti-H3 (Abcam, #ab1791), anti-H3K27me3 (Millipore, #07-

449), H2AK119ub (Cell Signaling Technology, #8240), anti-H3K36me3 (Abcam, #ab9050). PolII antibodies were as in[64]. All ChIP experiments were followed by qPCR with indicated primer pairs (Supplementary Table 2).

**Statistical analysis**
Statistical analyses were performed in Microsoft Excel and R Studio (R v4.0.2). *P* value, *P* value levels, and sample number are included in figures and figure legends. Significance calculations with Student's *t*-test were precluded by *F*-test to determine the usage of heteroscedastic or homoscedastic test. Unless otherwise stated, Student's *t*-tests were two-tail. *P* value levels were marked by asterisks on the figures as follows: * ($P \leq 0.05$), ** ($P \leq 0.01$), *** ($P \leq 0.001$).

**Reporting summary**
Further information on research design is available in the Nature Research Reporting Summary linked to this article.

## Data availability
The data that support this study are available from the corresponding authors upon reasonable request. Raw and processed proteomic data from VAL1 mass spectrometry was deposited to the ProteomeXchange Consortium (http://proteomecentral.proteomexchange.org) via PRIDE partner repository[65] and are available under dataset identifier: PXD036548.

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

## Acknowledgements

This work was funded by the BBSRC grant BB/P006590/1, Wellcome Trust grant 210654/Z/18/Z and supported by the BBSRC Institute Strategic Programmes GRO (BB/J004588/1) and GEN (BB/P013511/1) and a Royal Society Professorship to CD. The authors would like to thank all members of the Dean lab for discussions.

## Author contributions

P.M., P.W., and C.D. conceived the study. P.M. performed: yeast two-hybrid, FAIRE, nucleosome mobility, ChIPs and expression analyses in *val1* and PRC1 mutants. P.W. performed: expression analyses, histone marks and PolII ChIPs for ASAP mutants. D.Z. performed nucleosome position experiment. T.L. performed protein co-immunoprecipitation with mass spectrometry. T.L., G.S., and C.M. did proteomic data analysis. M.N. performed H2Aub ChIP in Supplementary Fig. 5. J.I.Q. initiated the genetic analysis. E.F.E. performed CoIP in Supplementary Fig. 2. P.M. and C.D. designed the experiments. C.D. lead the general concept of research. P.M. and C.D. wrote the paper with other authors contributing to their relevant sections of the manuscript. C.D. is the lead contact for this paper.

## Competing interests

The authors declare no competing interests.
