## [Peer Review File · Nature Communications]

VAL1 as an assembly platform co-ordinating co-transcriptional repression and chromatin regulation at Arabidopsis FLC.REVIEWER COMMENTS

Reviewer #1 (Remarks to the Author):

FLC is a central gene determining vernalization in Arabidopsis, and VAL1 serves as a critical transcriptional repressor for FLC expression during this process. In this manuscript entitled "VAL1 as an assembly platform co-ordinating co-transcriptional repression and chromatin regulation at Arabidopsis FLC" by Mikulski et al., the authors further dissected the detailed molecular mechanism of VAL1 in regulating FLC silencing. They first demonstrated that ASAP complex had a physical interaction with VAL1 via SAP18, and was involved in FLC silencing by various chromatin regulations at the nucleation region of FLC. Next, the authors verified that the VAL1 was associated with PRC1 complex and NDX involving nucleosome dynamics, and accumulation of H2Aub at the FLC. Therefore, the authors claimed that the VAL1 co-ordinated ASAP and PRC1 complexes leading to FLC silencing during vernalization. This is a potentially interesting story which does bring us some new clues for the mechanism of FLC-mediated vernalization in Arabidopsis. However, the authors should address following concerns carefully to support their conclusions.

Major points

1. Interactions of VAL1 with ASAP or PRC1 are very critical for the proposal of working model. However, the evidence based on the present data for the interaction is rather weak. It is better to perform BiFC, Pull-down or Co-IP assays to enhance the physical interactions among proteins.
2. The authors investigated a variety of chromatin features at the FLC locus in the mutants of ASAP components. According to several recent studies, the VAL proteins in Arabidopsis and rice affect deacetylation of target genes. The reviewer wonders whether the acetylation status of FLC locus is affected in the asap mutants.

The authors claimed a major effect of the asap mutants in both FRI and fri genotypes (lines 87-88), but the asap mutants in fri genotype did not mention and show up in any figure. Moreover, the authors claimed that VAL1 affects nucleosome stabilization at the nucleation region of FLC. However, no significant difference in chromatin accessibilities of Col FRI and val1 FRI was observed (Fig 3d, lines 135-138). The authors need to explain the contradictory results.

3. The authors should demonstrate the flowering phenotypes of mutants to justify the functions of ASAP and PRC1 complexes on FLC-mediated vernalization.

4. The authors described the function of ASAP and PRC1 complexes independently, and proposed that VAL1 co-ordinates co-transcriptional repression and chromatin regulation at the FLC by recruiting two different repressing complexes. However, the authors failed in providing any evidence to answer whether these two complexes function simultaneously or in turn.

5. The manuscript appeared rather rough in current version. The authors should improve the quality of data presentation. For example, the data showed in Fig S1 and Fig S7 were apparently derived from different pictures, a space was required for presentation to avoid misconduct; the Fig 5c was the same result of Fig 5b with different form, which can be deleted to avoid redundancy; legends for Fig. 1 and Fig 6 were inconsistent with the figures. To simplify the model, the reviewer suggests put VAL1 in the middle and share with two complexes in Fig 7.

Some figure titles were misleading such as Fig.5. VAL1, BMI1B and NDX effect on FLC expression, which also included RING1A; Fig.7 Functional relationship of ASAP, PRC1 and PRC2 in FLC regulation, which did not show PRC2 at all.

Minor points

1. The spelling mistakes should be corrected. The title may miss an "acts" after VAL1; "H2K27me3" in the Abstract (line 24) should be "H3K27me3".
2. Different forms for yeast-2-hybrid, yeast two hybrid, yeast-two-hybrid in the manuscript should be unified as yeast two-hybrid.
3. The writing quality should be improved by careful editing. P value and latin names should be italic; formats of the References should be consistent. The authors should pay more attention to the molecular formulas in the Materials and Methods section.

Reviewer #2 (Remarks to the Author):

Mikulski and colleagues report on their study about the function of VAL1 interaction with ASAP and PRC1 complexes in Arabidopsis. Compared to several previously published studies, here they provide new information about H2Aub level changes at FLC in different mutants and during vernalization and after return to warm temperature. They found interestingly that PRC1-mediated H2Aub accumulates only at the FLC nucleation region upon cold treatment and the H2Aub level is only transiently maintained after transfer back to warm. This H2Aub pattern is in contrast to that of H3K27me3, which spreads across the FLC locus and maintained after transfer back to warm.

Introduction

This study extends the previous study published in Science on 2016 by the same laboratory on VAL1 triggered Polycomb silencing at FLC during vernalization. Meanwhile several additional publications report physical/functional interactions of VAL1 with the PRC1 AtBMI1, LHP1 and the PRC2 CLF and SWN. These need to be elaborated and included in Introduction in order to more precisely appreciate the Results at the next section.

Results

Fig. 1 The legend does not fully correspond to the graph: a) inversed with c), and b) inversed with d). In addition, the cold-treatment time-course graphs of the FLC expression does not seem to support the conclusion 'the major effect of the asap mutants in both FRI and fri genotypes was in warm (NV) conditions' drawn in the body text. The small differences shown in sr45 FRI need to be assessed with statistics to verify significance. The yeast-two-hybrid data are not very convincing.

Fig 2. Better to show all graphs in one column to appreciate at the same position of amplicon. It would be interesting to also include H2Aub.

Fig. 3. The labeling n.s. should be described. It could be misleading, e.g. the differences between position '183' and position '-50' seem significant. This would against n.s. applying to all of column.

Fig 4. The H2Aub level in Col-0 also shows a peak around position '5600'. Is this region corresponding to the 3'-end of FLC or to a neighbor gene? If at 3'-end of FLC, any expected function?

Fig 5. The analysis of bmi1b and ring1A is barely informative. It is well-known that BMI1B is functionally redundant with BMI1A and RING1A is functionally redundant with RING1B.

Fig 6. The yeast-two-hybrid data are not very convincing (strong background). The H2Aub level in Col-0 around position '5600' is different between the two graphs shown in d). Any explanation?

Discussion

Fig 7. Previously published data on VAL1-PRC2 should be included in the model and more clearly discussed in the main text.

We are very grateful to the two reviewers for their excellent comments. We have undertaken the requested experiments and modified the documents according to all their comments. I am sorry this is a slightly delayed return – between individuals moving on to other labs and Covid it has been hard to complete these changes.

We provide a point-by-point response to the reviewers' comments below. We also have highlighted the main changes in the manuscript text file. The changes to the figures are as follows:

In this revised version, **Fig. 1** only includes data of the proteomic analysis of VAL1-HA. Fig. 1a is a new representation of LC-MS data. Fig. 1b corresponds to old Fig. 6a, with the addition of data for ASAP components.

Fig. 2 includes data for *FLC* expression in ASAP mutants. Fig. 2a corresponds to previous Fig. 1a, Fig. 2b to old Fig. 1b and Fig. 2d to old Fig. 1e. Fig. 2c was added in this revision to represent *FLC* expression levels in *sr45 FRI* vs *fri* background. Fig. 2e represents *FLC* gene model schematic with scale to aid in navigating through amplicon positions in Fig. 2-6.

Fig. 3 corresponds to old Fig. 2.

Fig. 4 corresponds to old Fig. 3.

Fig. 5 corresponds to old Fig. 4.

Data in **Fig. 6** show the requirement of VAL1 and NDX for accumulation of PRC1-dependent histone modification H2Aub at *FLC* locus. This new figure is a merge from old Fig. 5 and Fig. 6. Some data have been moved to the supplementary file. Fig. 6a is old Fig. 5a; Fig. 6b is old Fig. 6d; Fig. 6c is old Fig. 6e.

Fig. 7 depicts the model of functional relationship of ASAP/VAL/PRC1/PRC2 in *FLC* regulation. The model has been improved based on reviewer's comments.

Supplementary Fig. 1 represents the STRING analysis of VAL1-HA proteomics data. This is a new addition in this revised version.

We have moved all the yeast two-hybrid data to **Supplementary Fig. 2**. Supplementary Fig. 2a was previously Fig. 1c/Supplementary Fig. 1b; Supplementary Fig. 2c was old Fig. 6b and Supplementary Fig. 7a; Supplementary Fig. 2d was old Supplementary Fig. 7b. Supplementary Fig. 2b depicts the coIP experiments in HEK293 cells that were conducted during this revision.

Supplementary Fig. 3 includes the phenotypic characterization of ASAP mutants. Flowering data of Supplementary Fig. 3a were newly added during the revision of this manuscript. Supplementary Fig. 3b was old Supplementary Fig. 1a; Supplementary Fig. 3c was newly added in this revision; Supplementary Fig. 3d was Supplementary Fig. 2.

Supplementary Fig. 4 is old Supplementary Fig. 3.

The H2Aub dynamics during longer cold included represented in **Supplementary Fig. 5** is new to this revised version of the manuscript.

Supplementary Fig. 6 was old Supplementary Fig. 10.

Supplementary Fig. 7 shows data of *FLC* and *COOLAIR* expression in *val1/PRC1* mutants. Supplementary Fig. 7a was old Fig. 5b; Supplementary Fig. 7b was old Fig. 5c; Supplementary Fig. 7c was old Supplementary Fig. 5c; Supplementary Fig. 7d was old Supplementary Fig. 6.

Supplementary Fig. 8 corresponds to old Supplementary Fig. 4 and includes newly added flowering data for PRC1 mutants in *FRI+* background (Supplementary Fig. 8c).

Supplementary Fig. 9 corresponds to old Supplementary Fig. 8 (now Supplementary Fig. 9a) and Supplementary Fig. 9 (now Supplementary Fig. 9b).

Following a comment from Reviewer #1 about redundancy with previously published data, we have decided to remove old Supplementary Fig. 5a-b, which was showing *FLC* expression in *val1* and *ring1a/b* mutants in Col-0 background (*fri*).

Reviewer #1 (Remarks to the Author):

FLC is a central gene determining vernalization in Arabidopsis, and VAL1 serves as a critical transcriptional repressor for *FLC* expression during this process. In this manuscript entitled “VAL1 as an assembly platform co-ordinating co-transcriptional repression and chromatin regulation at Arabidopsis *FLC*” by Mikulski et al., the authors further dissected the detailed molecular mechanism of VAL1 in regulating *FLC* silencing. They first demonstrated that ASAP complex had a physical interaction with VAL1 via SAP18, and was involved in *FLC* silencing by various chromatin regulations at the nucleation region of *FLC*. Next, the authors verified that the VAL1 was associated with PRC1 complex and NDX involving nucleosome dynamics, and accumulation of H2Aub at the *FLC*. Therefore, the authors claimed that the VAL1 co-ordinated ASAP and PRC1 complexes leading to *FLC* silencing during vernalization. This is a potentially interesting story which does bring us some new clues for the mechanism of *FLC*-mediated vernalization in Arabidopsis. However, the authors should address following concerns carefully to support their conclusions.

Major points

1. Interactions of VAL1 with ASAP or PRC1 are very critical for the proposal of working model. However, the evidence based on the present data for the interaction is rather weak. It is better to perform BiFC, Pull-down or Co-IP assays to enhance the physical interactions among proteins.

Part of the delay in returning our revised manuscript was our difficulty in expressing VAL fusions transiently in *N. benthamiana* plants to undertake the requested assays. We therefore turned to Co-IP experiments in transfected mammalian cells and have included these data in Supplementary Fig. 2b and explanation in the text (#lines 104-105). We also supplement the revised version with extensive in vivo proteomics data from plants at different stages of the vernalization process (Fig. 1 and Supplementary Fig. 1). Overall, these new data strengthen the view that there are few direct interactions of sufficient affinity that the proteins co-IP, but that the system is built around multivalent interactions that mutually reinforce association of the larger complexes.

2. The authors investigated a variety of chromatin features at the *FLC* locus in the mutants of ASAP components. According to several recent studies, the VAL proteins in Arabidopsis and rice affect deacetylation of target genes. The reviewer wonders whether the acetylation status of *FLC* locus is affected in the asap mutants.

As pointed by the reviewer, histone acetylation activity has been linked to VAL1 and the ASAP component SAP18 by previous reports^{1,2}. In line with this, we have previously demonstrated that H3 acetylation fails to be removed from *FLC* chromatin during vernalization in *val1* mutants, which correlated with increased *FLC* transcript levels³. All our previous analyses have shown that histone acetylation directly correlates with transcription level at *FLC*. This is the reason why we have not analysed the acetylation status of *FLC* in ASAP mutants as it would reflect a confounding effect from increase in transcription itself.

The authors declaimed a major effect of the asap mutants in both *FRI* and *fri* genotypes (lines 87- 88), but the asap mutants in *fri* genotype did not mention and show up in any figure.

Regarding *FRI* vs *fri* in ASAP mutants, we have edited the text and figures to clarify which genotype is being used in each experiment. Fig. 2a shows *FLC* expression in ASAP *FRI* mutants. Fig. 2b shows *FLC* expression in *val1* and ASAP *fri* (single and double) mutants, highlighted in lines #113-123. In this revised version, we have added a new Fig. 2c which shows a very interesting behaviour of *sr45*: the differential effect of *sr45* in *fri* vs *FRI+* background; highlighted in lines #134-137. Fig. 2d, Fig. 3 and Supplementary Fig. 3a, b and d present data for ASAP *fri* mutants. Supplementary Fig. 3c presents data for ASAP *FRI* mutants in NV.

Moreover, the authors claimed that VAL1 affects nucleosome stabilization at the nucleation region of *FLC*. However, no significant difference in chromatin accessibilities of Col*FRI* and *val1FRI* was observed (Fig 3d, lines 135-138). The authors need to explain the contradictory results.

Thank you for this comment, we agree that our findings needed further explanation. Our data (now in Fig. 4) show that VAL1 facilitates nucleosome stabilization at specific positions, as assayed by a low-salt solubility assay. This assay has been used in other studies to analyse nucleosome dynamics during torsional stress in chromatin⁴. The observed VAL-dependent changes are not reflected in changes in the degree of general accessibility of the 5' end of *FLC* as judged by FAIRE⁵. Current work in the field is dissecting the biophysical nature of nucleosome dynamics – which are clearly not the stable structures commonly drawn in text-books. We interpret the observed changes in low-salt solubility as showing subtle but functionally important changes in nucleosome behaviour important for transcriptional repression – but not representing gross changes in nucleosome occupancy⁵. We have clarified this in lines #167-169 and #178-181.

3. The authors should demonstrate the flowering phenotypes of mutants to justify the functions of ASAP and PRC1 complexes on *FLC*-mediated vernalization.

In this revised version, we have edited text and figures to clarify the distinct functions of PRC1 and ASAP in different aspects of *FLC* silencing. Our data suggest that despite higher *FLC* transcript levels in ASAP mutants in the warm (Fig. 2b), the ASAP complex is not required for the Arabidopsis vernalization response (mechanistic redundancy potentially explaining this). We have not detected significant effects of *FLC* silencing in ASAP mutants during vernalization (Fig. 2a), both considering rate of downregulation during cold and *FLC* post-cold levels. The lack of vernalization phenotypes for ASAP mutants (in *FRI+*) is also reflected in the proteomics data (Fig. 1), where the association of ASAP/VAL1-HA observed in the warm is lost upon cold exposure. Thus, we conclude that the major effect of ASAP on *FLC* regulation takes place in the warm (non-vernalization conditions). ASAP is involved in co-transcriptional repression of *FLC* in the warm (Fig. 3). Interestingly, despite the higher *FLC* expression levels in ASAP mutants, we have not detected a significant delay in flowering time in the warm (Supplementary Fig. 3a), despite a minor effect in *acinus* mutant (see lines #119-123). Given the role of ASAP in co-transcriptional gene repression, one possibility would be that *FLC* downstream targets (floral integrator genes) are also upregulated in *sap18* and *sr45* mutants, and thus flowering is not affected.

Regarding PRC1, our data support that accumulation of H2Aub (PRC1 prominent mark) at *FLC* PHD-PRC2 nucleation region correlates with transcriptional shutdown of *FLC* during cold, and acts as an intermediate step towards the more slowly accumulating H3K27me3. We demonstrate that the H2Aub peak at *FLC* is dependent on VAL1 binding to *FLC* chromatin. Therefore, we propose a model in which VAL1 provides a platform to recruit a range of co-transcriptional regulators including PRC1 that efficiently silence *FLC* during vernalization. A deficiency in PRC1 function (PRC1 *FRI*

mutants) should negatively affect the floral transition after vernalization, as observed for *val1 FRI* mutants³. However, given the redundancy between PRC1 components, we only observe a slight delay in flowering time in *ring1A FRI* mutants after 4W cold (Supplementary Fig. 8c; lines #232-235) and no delay in *bmi1A FRI* and *bmi1B FRI* when compared to WT (*Col FRI*). Our observations are in line with reports in the literature. *RING1A* promotes flowering by repressing expression of *MAF4* and *MAF5*⁶. The same work reported that the PRC1 single mutants *ring1b*, *bmi1a* and *bmi1b*, do not influence flowering time. Similarly, *bmi1c* single mutant flowers at the same time as WT⁷. Multiple mutant combinations (*ring1a/ring1b* and *bmi1a/bmi1b/bmi1c*) lead to very strong developmental phenotypes with seedlings aborting growth soon after germination⁸⁻¹⁰.

4. The authors described the function of ASAP and PRC1 complexes independently, and proposed that VAL1 co-ordinates co-transcriptional repression and chromatin regulation at the *FLC* by recruiting two different repressing complexes. However, the authors failed in providing any evidence to answer whether these two complexes function simultaneously or in turn.

Thank you for this constructive comment. We have hopefully improved the text and figures describing this part of the work trying to clarify the distinct functions of PRC1 and ASAP at different phases of *FLC* silencing, in both cases dependent on VAL1. In Fig. 1, our proteomics analysis shows distinct representation of ASAP and PRC1 components in warm and cold, respectively. We have demonstrated that ASAP/VAL1 complex is required to set *FLC* transcript levels in the warm (Fig. 2b-d), likely through modulating PolII enrichment (Fig. 3a) and repressive histone accumulation at the locus (Fig. 3b). In contrast, ASAP seems not to be involved in *FLC* silencing during vernalization (Fig. 2a). In the case of PRC1, this work demonstrates for the first time the dynamic accumulation of H2Aub at *FLC* nucleation region during increasing weeks of cold (Fig. 5a), a process that is VAL1- and NDX-dependent (Fig. 6b-c). Interestingly, and unlike H3K27me3, H2Aub peak is transient (Supplementary Fig. 5) and does not spread beyond the nucleation region (Fig. 5c). Therefore, we propose a more prominent role of PRC1 in the cold and of ASAP before vernalization. However, from our data, we cannot exclude the possibility that PRC1 and ASAP complexes may function together at other stages of *FLC* silencing that we have not been able to capture under our working conditions.

5. The manuscript appeared rather rough in current version. The authors should improve the quality of data presentation.

Thank you for the comments. We have revised the text and figures to improve the manuscript and clarify the points raised. We have detailed above all the changes in figures.

For example, the data showed in Fig S1 and Fig S7 were apparently derived from different pictures, a space was required for presentation to avoid misconduct;

We have moved all the yeast two-hybrid data to Supplementary Fig. 2. Supplementary Fig. 2a (old Supplementary Fig. 1b) and Supplementary Fig. 2b,c (old Supplementary Fig. 7) are separate experiments. Supplementary Fig. 2a is for VAL1-SAP18 and Supplementary Fig. 2b,c are VAL1-NDX & NDX-RING1A/RING1B.

the Fig 5c was the same result of Fig 5b with different form, which can be deleted to avoid redundancy;

The mentioned figures (old Fig. 5b and 5c) have been moved to supplementary files (now Supplementary Fig. 7a,b). Supplementary Fig. 7b and c shows fold to NV, while Supplementary Fig. 7a shows absolute expression. We believe that it is important to keep both figures to show that *FLC* levels are upregulated in *val1* and *val1 bmi1b* already in NV starting conditions.

legends for Fig. 1 and Fig 6 were inconsistent with the figures.

As detailed above, Fig. 1 and Fig. 6 have been modified. Legends have been edited accordingly.

To simplify the model, the reviewer suggests put VAL1 in the middle and share with two complexes in Fig 7.

We have edited the model of Fig. 7 to better clarify that VAL1 is part of both PRC1 and ASAP complex.

Some figure titles were misleading such as Fig.5. VAL1, BMI1B and NDX effect on *FLC* expression, which also included RING1A;

Sorry for the inconvenience. Old Fig. 5 has substantially changed and is now Fig. 6. The figure title has been changed to “**VAL1, PRC1 and NDX are required for H2Aub accumulation at the *FLC* nucleation region**”.

Fig.7 Functional relationship of ASAP, PRC1 and PRC2 in *FLC* regulation, which did not show PRC2 at all.

H3K27me3 and PRC1-PRC2 interplay have been added to the model in Fig. 7 (Fig. 7b).

Minor points

1. The spelling mistakes should be corrected. The title may miss an “acts” after VAL1; “H2K27me3” in the Abstract (line 24) should be “H3K27me3”.

Thank you for the suggestion. We have included “acts” in the title and corrected the misspelling in the abstract.

2. Different forms for yeast-2-hybrid, yeast two hybrid, yeast-two-hybrid in the manuscript should be unified as yeast two-hybrid.

We have included the suggested modification.

3. The writing quality should be improved by careful editing. P value and latin names should be italic; formats of the References should be consistent. The authors should pay more attention to the molecular formulas in the Materials and Methods section.

Thank you for the useful comments. We have improved the writing following your suggestions: p-value changed to *P* value; $p < 0.0xxx$ changed to $P < 0.0xxx$ in figure legends and text; latin names (*bona fide*) italicized; molecular formulas (CaCl_2 and MgCl_2) changed to subscript 2.

Reviewer #2 (Remarks to the Author):

Mikulski and colleagues report on their study about the function of VAL1 interaction with ASAP and PRC1 complexes in Arabidopsis. Compared to several previously published studies, here they provide new information about H2Aub level changes at *FLC* in different mutants and during vernalization and after return to warm temperature. They found interestingly that PRC1-mediated H2Aub accumulates only at the *FLC* nucleation region upon cold treatment and the H2Aub level is only transiently maintained after transfer back to warm. This H2Aub pattern is in contrast to that of H3K27me3, which spreads across the *FLC* locus and maintained after transfer back to warm.

Introduction

This study extends the previous study published in Science on 2016 by the same laboratory on VAL1 triggered Polycomb silencing at *FLC* during vernalization. Meanwhile several additional publications report physical/functional interactions of VAL1 with the PRC1 AtBMI1, LHP1 and the PRC2 CLF and SWN. These need to be elaborated and included in Introduction in order to more precisely appreciate the Results at the next section.

Following the reviewer's suggestion, we have extended the Introduction to include reported VAL1 interactors (lines #66-71). In addition, we have also cited the interaction with LHP1 and other partners in the Discussion (lines #294-296).

Results

Fig. 1 The legend does not fully correspond to the graph: a) inversed with c), and b) inversed with d).

These data (old Fig. 1) are now presented in Fig. 2 and Supplementary Fig. 2. We have corrected the mistakes in the legend.

In addition, the cold-treatment time-course graphs of the *FLC* expression does not seem to support the conclusion 'the major effect of the asap mutants in both *FRI* and *fri* genotypes was in warm (NV) conditions' drawn in the body text.

We have edited the text to clarify the effects observed in ASAP mutants in *FRI* vs *fri* background (results section, lines #113-123 and #134-137). The cold-treatment time-course is now presented in Fig. 2a, *FLC* expression levels in NV in *val1*/ASAP mutants are in Fig. 2b and we have added one additional figure (Fig. 2c) to show the differential effect of *FRI* vs *fri* in *sr45* mutant background. Please also refer to our response to Reviewer 1 (point 3).

The small differences shown in *sr45 FRI* need to be assessed with statistics to verify significance. The yeast-two-hybrid data are not very convincing.

We have added statistics to Fig. 2a (old Fig. 1a), which show not significant effect of ASAP on *FLC* silencing during vernalization. To support the yeast two-hybrid data, we have supplemented this work with co-IP analysis in transfected mammalian HEK cells (Supplementary Fig. 2b).

Fig 2. Better to show all graphs in one column to appreciate at the same position of amplicon. It would be interesting to also include H2Aub.

We have edited this figure (now Fig. 3) as suggested. Given the lack of evidence of a link between PRC1 and ASAP in the warm, we did not investigate the accumulation of H2Aub in ASAP mutants.

Fig. 3. The labeling n.s. should be described. It could be misleading, e.g. the differences between position '183' and position '-50' seem significant. This would against n.s. applying to all of column. We have edited this figure (now Fig. 4d) and its legend to clarify the meaning of n.s. We have added one sentence in the legend explaining how the statistics were calculated.

Fig 4. The H2Aub level in Col-0 also shows a peak around position '5600'. Is this region corresponding to the 3'-end of *FLC* or to a neighbor gene? If at 3'-end of *FLC*, any expected function?

These data are now in Fig. 5. The 5600 position corresponds to *FLC* genomic region, around exon6/exon7. Interestingly, the H2Aub peak is only observed in Col-0 *fri*, in which *FLC* gene is fully repressed. This could be explained by the either presence of a previously described gene loop¹¹ that

connects *FLC* 5' and 3' end, or the effect of antisense transcription. We have included this explanation in the main text (lines #201-203, #246-247 and #252-254).

Fig 5. The analysis of *bmi1b* and *ring1A* is barely informative. It is well-known that *BMI1B* is functionally redundant with *BMI1A* and *RING1A* is functionally redundant with *RING1B*.

Thank you for your comment. Indeed, previous reports have shown functional redundancy of PRC1 components. However, all previous analyses were done in Col-0 *fri* background. In this manuscript, we investigated the vernalization phenotype of PRC1 mutants in *FRI+* (winter requirement background), as well as *val1*/PRC1 mutant combinations. Anyway, except for *ring1A FRI*, functional redundancy prevents us from fully defining a vernalization phenotype for PRC1 *FRI* mutant plants in the cold and post-cold.

Fig 6. The yeast-two-hybrid data are not very convincing (strong background). Indeed, the *SAP18-VAL1* yeast two-hybrid interaction is relatively weak. We have supplemented this work with co-IP analysis in transfected mammalian HEK cells, which shows these proteins do not interact sufficiently strongly for co-immunoprecipitation. We edited the manuscript accordingly.

The H2Aub level in Col-0 around position '5600' is different between the two graphs shown in d). Any explanation?

H2Aub ChIP experiment in Col-0 and *ndx* (now Fig. 6b, right panel) did not include 5600 amplicon, whereas H2Aub ChIP in Col-0 and *val1* (now Fig. 6b, left panel) did. Originally, H2Aub ChIP in Col-0/*ndx* has been performed prior to Col-0/*val1* experiment, without the knowledge about interesting H2Aub enrichment around this region. Please see also our reply to the comment to Fig.4 above. We believe H2Aub enrichment at this region might potentially correspond to 5'-3' end gene loop at *FLC* described previously¹¹ or might be related to *COOLAIR* R-loop at *FLC* 3' end^{12,13}. We have added relevant sentences to the main text (lines #201-203, #246-247 and #252-254).

Discussion

Fig 7. Previously published data on *VAL1*-PRC2 should be included in the model and more clearly discussed in the main text.

Thank you for your comment. We agree with the reviewer that PRC2 components (*SWN*/*CLF*) play an important role in *VAL1*-dependent *FLC* regulation^{3,14,15}. However, our proteomic analysis did not show evidence of the direct interaction of *VAL1*-PRC2 suggested by Yuan et al., 2021¹⁵. Rather, our work suggests an indirect and more complex relationship, where PRC1 remodels *FLC* chromatin promoting PRC2 function. Our model is in line with conclusions put forward before (i.e. Baile et al., 2021¹⁴ and Zhou et al., 2017¹⁶). We have included a simplified version of the interplay between PRC1-PRC2 in the model (Fig. 7). The previously described links between *VAL1* and PRC2 (*LHP1*, *CLF*, *SWN*) are now mentioned in the introduction (lines #66-71 and #74-77) and discussion (lines #294-296).

Kind regards,
Manuscript authors

References (used in this document):

1. Zeng, X. et al. HISTONE DEACETYLASE 9 functions with Polycomb silencing to repress *FLOWERING LOCUS C* expression. *Plant Physiol.* **182**, 555-565 (2020).

2. Zhang, Y., Iratni, R., Erdjument-Bromage, H., Tempst, P. & Reinberg, D. Histone deacetylases and SAP18, a novel polypeptide, are components of a human Sin3 complex. *Cell* **89**, 357-364 (1997).
3. Qüesta, J. I. *et al.* Arabidopsis transcriptional repressor VAL1 triggers Polycomb silencing at *FLC* during vernalization. *Science* **353**, 485–488 (2016).
4. Teves, S. S. & Henikoff, S. DNA torsion as a feedback mediator of transcription and chromatin dynamics. *Nucl. (United States)* **5**, 211-218 (2014).
5. Henikoff, S., Henikoff, J. G., Sakai, A., Loeb, G. B. & Ahmad, K. Genome-wide profiling of salt fractions maps physical properties of chromatin. *Genome Res.* **19**, 460–469 (2009).
6. Shen, L. *et al.* The putative PRC1 RING-finger protein AtRING1A regulates flowering through repressing MADS AFFECTING FLOWERING genes in Arabidopsis. *Dev.* **141**, 1303-1312 (2014).
7. Li, W. *et al.* Overexpression of AtBMI1C, a Polycomb Group protein gene, accelerates flowering in Arabidopsis. *PLoS One* **6**, e21364 (2011).
8. Yang, C. *et al.* VAL-and AtBMI1-Mediated H2Aub initiate the switch from embryonic to postgerminative growth in arabidopsis. *Curr. Biol.* **23**, 1324–1329 (2013).
9. Bratzel, F., López-Torrejón, G., Koch, M., Del Pozo, J. C. & Calonje, M. Keeping cell identity in arabidopsis requires PRC1 RING-finger homologs that catalyze H2A monoubiquitination. *Curr. Biol.* **20**, 1853–1859 (2010).
10. Chen, D., Molitor, A., Liu, C. & Shen, W. H. The Arabidopsis PRC1-like RING-finger proteins are necessary for repression of embryonic traits during vegetative growth. *Cell Res.* **20**, 1332–1344 (2010).
11. Crevillén, P., Sonmez, C., Wu, Z. & Dean, C. A gene loop containing the floral repressor *FLC* is disrupted in the early phase of vernalization. *EMBO J.* **32**, 140–148 (2013).
12. Sun, Q., Csorba, T., Skourti-Stathaki, K., Proudfoot, N. J. & Dean, C. R-loop stabilization represses antisense transcription at the Arabidopsis *FLC* locus. *Science* **340**, 619–621 (2013).
13. Xu, C. *et al.* R-loop resolution promotes co-transcriptional chromatin silencing. *Nat. Commun.* **12**, 1790 (2021).
14. Baile, F., Merini, W., Hidalgo, I. & Calonje, M. EAR domain-containing transcription factors trigger PRC2-mediated chromatin marking in Arabidopsis. *Plant Cell*, koab139 (2021).
15. Yuan, L. *et al.* The transcriptional repressors VAL1 and VAL2 recruit PRC2 for genome-wide Polycomb silencing in Arabidopsis. *Nucleic Acids Res.* **49**, 98–113 (2021).
16. Zhou, Y., Romero-Campero, F. J., Gómez-Zambrano, Á., Turck, F. & Calonje, M. H2A monoubiquitination in *Arabidopsis thaliana* is generally independent of LHP1 and PRC2 activity. *Genome Biol.* **18**, 69 (2017).

REVIEWERS' COMMENTS

Reviewer #1 (Remarks to the Author):

The authors have addressed most of my concerns. I support the publication of the revised manuscript.

Reviewer #2 (Remarks to the Author):

In this revised version, the authors added new data and carefully revised throughout the manuscript. All my points raised in the previous version have been addressed to satisfaction. I don't have additional comments/questions, excepting the spelling use of NDX1 or NDX should be fixed throughout the manuscript.

Please find our response (marked in blue) to the reviewer comment below:

REVIEWERS' COMMENTS (create separate file for this section)

Reviewer #1 (Remarks to the Author):

The authors have addressed most of my concerns. I support the publication of the revised manuscript.

Reviewer #2 (Remarks to the Author):

In this revised version, the authors added new data and carefully revised throughout the manuscript. All my points raised in the previous version have been addressed to satisfaction. I don't have additional comments/questions, excepting the spelling use of NDX1 or NDX should be fixed throughout the manuscript.

Thank you for your comments. We have corrected the NDX/NDX1 spelling. Following previous publications (Sun et al Science 2013, doi:10.1126/science.1234848; Zhu et al Plant Cell 2020, doi:10.1105/tpc.19.00604), we use NDX and *ndx* to refer to the protein and the mutant line (*ndx1-4* allele), respectively.